# DEFT: Dexterous Fine-Tuning for Hand Policies

**Aditya Kannan**[*]    **Kenneth Shaw**[*]    **Shikhar Bahl**
**Pragna Mannam**    **Deepak Pathak**

Carnegie Mellon University

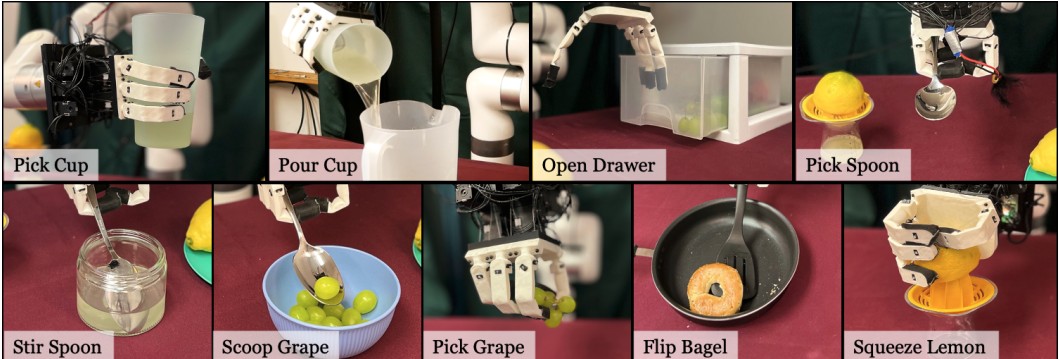

Figure 1: We present DEFT, a novel approach that can learn complex, dexterous tasks in the real world in an efficient manner. DEFT manipulates tools and soft objects without any robot demonstrations.

**Abstract:** Dexterity is often seen as a cornerstone of complex manipulation. Humans are able to perform a host of skills with their hands, from making food to operating tools. In this paper, we investigate these challenges, especially in the case of soft, deformable objects as well as complex, relatively long-horizon tasks. However, learning such behaviors from scratch can be data inefficient. To circumvent this, we propose a novel approach, DEFT (**DE**xterous **F**ine-**T**uning for Hand Policies), that leverages human-driven priors, which are executed directly in the real world. In order to *improve* upon these priors, DEFT involves an efficient online optimization procedure. With the integration of human-based learning and online fine-tuning, coupled with a soft robotic hand, DEFT demonstrates success across various tasks, establishing a robust, data-efficient pathway toward general dexterous manipulation. Please see our website at https://dexterous-finetuning.github.io for video results.

**Keywords:** Dexterous Manipulation, Learning from Videos

## 1 Introduction

The longstanding goal of robot learning is to build robust agents that can perform long-horizon tasks autonomously. This could for example mean a self-improving robot that can build furniture or an agent that can cook for us. A key aspect of most tasks that humans would like to perform is that they require complex motions that are often only achievable by hands, such as hammering a nail or using a screwdriver. Therefore, we investigate dexterous manipulation and its challenges in the real world.

A key challenge in deploying policies in the real world, especially with robotic hands, is that there exist many failure modes. Controlling a dexterous hand is much harder than end-effectors due to larger action spaces and complex dynamics. To address this, one option is to *improve* directly in the real world via *practice*. Traditionally, reinforcement learning (RL) and imitation learning (IL) techniques have been used to deploy hands-on tasks such as in-hand rotation or grasping. This is

---

[*]Equal contribution, order decided by coin flip.

7th Conference on Robot Learning (CoRL 2023), Atlanta, USA.

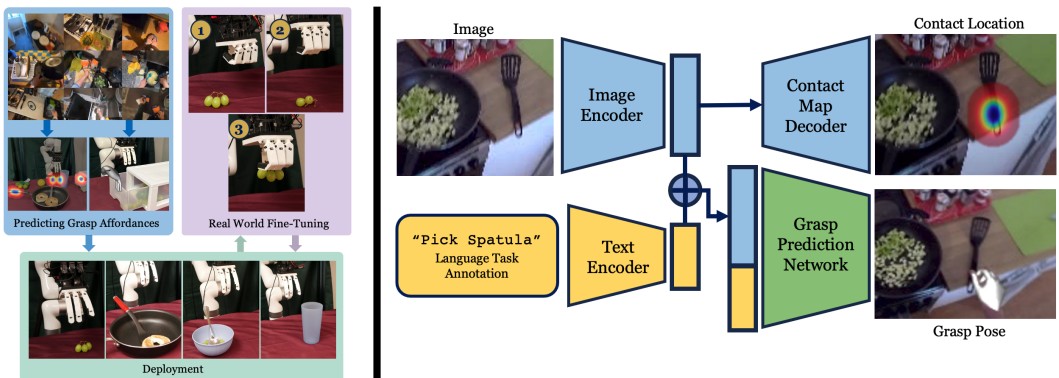

Figure 2: **Left:** DEFT consists of two phases: an affordance model that predicts grasp parameters followed by online fine-tuning with CEM. **Right:** Our affordance prediction setup predicts grasp location and pose.

the case as setups are often built so that it is either easy to simulate in the real world or robust to practice. However, the real world contains tasks that one cannot simulate (such as manipulation of soft objects like food) or difficult settings in which the robot cannot practice (sparse long-horizon tasks like assembly). How can we build an approach that can scale to such tasks?

There are several issues with current approaches for practice and improvement in the real world. Robot hardware often breaks, especially with the amount of contact to learn dexterous tasks like operating tools. We thus investigate using a *soft anthropomorphic hand* [1], which can easily run in the real world without failures or breaking. This soft anthropomorphic hand is well-suited to our approach as it is flexible and can gently handle object interactions. The hand does not get damaged by the environment and is robust to continuous data collection. Due to its human-like proportions and morphology, retargeting human hand grasps to robot hand grasps is made simpler.

Unfortunately, this hand is difficult to simulate due to its softness. Directly learning from scratch is also difficult as we would like to build *generalizable policies*, and not practice for every new setting. To achieve efficient real-world learning, we must learn a prior for reasonable behavior to explore using useful actions. Due to recent advances in computer vision, we propose *leveraging human data to learn priors* for dexterous tasks, and improving on such priors in the real world. We aim to use the vast corpus of internet data to define this prior. What is the best way to combine human priors with online practice, especially for hand-based tasks? When manipulating an object, the first thing one thinks about is where on the object to make contact, and how to make this contact. Then, we think about how to move our hands *after the contact*. In fact, this type of prior has been studied in computer vision and robotics literature as *visual affordances* [2, 3, 4, 5, 6, 7, 8, 9]. Our approach, DEFT, builds grasp affordances that predict the contact point, hand pose at contact, and post contact trajectory. To improve upon these, we introduce a sampling-based approach similar to the Cross-Entropy Method (CEM) to fine-tune the grasp parameters in the real world for a variety of tasks. By learning a residual policy [10, 11], CEM enables iterative real-world improvement in less than an hour.

In summary, our approach (DEFT) executes real-world learning on a soft robot hand with only a few trials in the real world. To facilitate this efficiently, we train priors on human motion from internet videos. We introduce 9 challenging tasks (as seen in Figure 1) that are difficult even for trained operators to perform. While our method begins to show good success on these tasks with real-world fine-tuning, more investigation is required to complete these tasks more effectively.

## 2 Related Work

**Real-world robot learning** Real-world manipulation tasks can involve a blend of classical and learning-based methods. Classical approaches like control methods or path planning often use hand-crafted features or objectives and can often lack flexibility in unstructured settings [12, 13, 14]. On the other hand, data-driven approaches such as deep reinforcement learning (RL) can facilitate complex behaviors in various settings, but these methods frequently rely on lots of data, privileged

reward information and struggle with sample efficiency [15, 16, 17, 18, 19]. Efforts have been made to scale end-to-end RL [20, 21, 22, 23, 24, 25] to the real world, but their approaches are not yet efficient enough for more complex tasks and action spaces and are reduced to mostly simple tasks even after a lot of real-world learning. Many approaches try to improve this efficiency such as by using different action spaces [26], goal relabeling [27], trajectory guidance [28], visual imagined goals [21], or curiosity-driven exploration [29]. Our work focuses on learning a prior from human videos in order to learn efficiently in the real world.

**Learning from Human Motion** The field of computer vision has seen much recent success in human and object interaction with deep neural networks. The human hand is often parametrized with MANO, a 45-dimensional vector [30] of axes aligned with the wrist and a 10-dimensional shape vector. MANOtorch from [31] aligns it with the anatomical joints. Many recent works detect MANO in monocular video [32, 33, 34]. Some also detect objects as well as the hand together [6, 35]. We use FrankMocap to detect the hand for this work. There are many recent datasets including the CMU Mocap Database [36] and Human3.6M [37] for human pose estimation, 100 Days of Hands [6] for hand-object interactions, FreiHand [38] for hand poses, Something-Something [39] for semantic interactions. ActivityNet datasets [40], or YouCook [41] are action-driven datasets that focus on dexterous manipulation. We use these three datasets: [42] is a large-scale dataset with human-object interactions, [43] for curated human-object interactions, and [44] which has many household kitchen tasks. In addition to learning exact human motion, many others focus on learning priors from human motion. [45, 46] learn general priors using contrastive learning on human datasets.

**Learning for Dexterous Manipulation** With recent data-driven machine learning methods, roboticists are now beginning to learn dexterous policies from human data as well. Using the motion of a human can be directly used to control robots [47, 48, 49]. Moving further, human motion in internet datasets can be retargeted and used directly to pre-train robotic policies [50, 51]. Additionally, using human motion as a prior for RL can help with learning skills that are human-like [52, 53, 54]. Without using human data as priors, object reorientation using RL has been recently successful in a variety of settings [55, 56]. Similar to work in robot dogs which do not have an easy human analog to learn from, these methods rely on significant training data from simulation with zero-shot transfer [57, 58].

**Soft Object Manipulation** Manipulating soft and delicate objects in a robot's environment has been a long-standing problem. Using the torque output on motors, either by measuring current or through torque sensors, is useful feedback to find out how much force a robot is applying [59, 60]. Coupled with dynamics controllers, these robots can learn not to apply too much torque to the environment around them [61, 62, 63]. A variety of touch sensors [64, 65, 66, 67] have also been developed to feel the environment around it and can be used as control feedback. Our work does not rely on touch sensors. Instead, we practice in the real world to learn stable and precise grasps.

## 3 Fine-Tuning Affordance for Dexterity

The goal of DEFT is to learn useful, dexterous manipulation in the real world that can generalize to many objects and scenarios. DEFT learns in the real world and fine-tunes robot hand-to-object interaction in the real world using only a few samples. However, without any priors on useful behavior, the robot will explore inefficiently. Especially with a high-dimensional robotic hand, we need a strong prior to effectively explore the real world. We thus train an affordance model on human videos that leverages human behavior to learn reasonable behaviors the robot should perform.

### 3.1 Learning grasping affordances

To learn from dexterous interaction in a sample efficient way, we use human hand motion as a prior for robot hand motion. We aim to answer the following: (1) What useful, actionable information can we extract from the human videos? (2) How can human motion be translated to the robot embodiment to guide the robot? In internet videos, humans frequently interact with a wide variety of objects. This

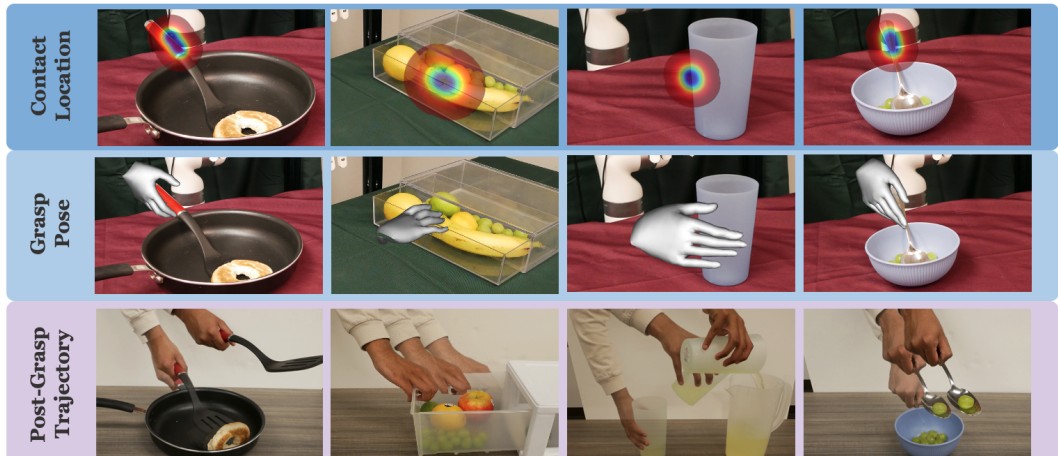

Figure 3: We produce three priors from human videos: the contact location (**top row**) and grasp pose (**middle row**) from the affordance prior; the post-grasp trajectory (**bottom row**) from a human demonstration of the task.

data is especially useful in learning object affordances. Furthermore, one of the major obstacles in manipulating objects with few samples is accurately grasping the object. A model that can perform a strong grasp must learn *where* and *how* to grasp. Additionally, the task objective is important in determining object affordances–humans often grasp objects in different ways depending on their goal. Therefore, we extract three items from human videos: the grasp location, human grasp pose, and task.

Given a video clip $V = \{v_1, v_2, \ldots, v_T\}$, the first frame $v_t$ where the hand touches the object is found using an off-the-shelf hand-object detection model [6]. Similar to previous approaches [3, 4, 7, 5], a set of contact points are extracted to fit a Gaussian Mixture Model (GMM) with centers $\mu = \{\mu_1, \mu_2, \ldots, \mu_k\}$. Detic [68] is used to obtain a cropped image $v'_1$ containing just the object in the initial frame $v_1$ to condition the model. We use Frankmocap [34] to extract the hand grasp pose $P$ in the contact frame $v_t$ as MANO parameters. We also obtain the wrist orientation $\theta_{\text{wrist}}$ in the camera frame. This guides our prior to output wrist rotations and hand joint angles that produce a stable grasp. Finally, we acquire a text description $T$ describing the action occurring in $V$.

We extract affordances from three large-scale, egocentric datasets: Ego4D [42] for its large scale and the variety of different scenarios depicted, HOI4D [8] for high-quality human-object interactions, and EPIC Kitchens [44] for its focus on kitchen tasks similar to our robot's. We learn a task-conditioned affordance model $f$ that produces $(\hat{\mu}, \hat{\theta}_{\text{wrist}}, \hat{P}) = f(v'_1, T)$. We predict $\hat{\mu}$ in similar fashion to [3]. First, we use a pre-trained visual model [69] to encode $v'_1$ into a latent vector $z_v$. Then we pass $z_v$ through a set of deconvolutional layers to get a heatmap and use a spatial softmax to estimate $\hat{\mu}$.

To determine $\hat{\theta}_{\text{wrist}}$ and $\hat{P}$, we use $z_v$ and an embedding of the text description $z_T = g(T)$, where $g$ is the CLIP text encoder [70]. Because transformers have seen success in encoding various multiple modes of input, we use a transformer encoder $\mathcal{T}$ to predict $\hat{\theta}_{\text{wrist}}, \hat{P} = \mathcal{T}(z_v, z_T)$. Overall, we train our model to optimize

| Parameter | Dimensions | Description |
|:---:|:---:|:---:|
| $\mu$ | 3 | XYZ grasp location in workspace |
| $\theta_{\text{wrist}}$ | 3 | Wrist grasp rotation (euler angles) |
| $P$ | 16 | Finger joint angles in soft hand |

Table 1: Parameters that are fine-tuned in the real world. The affordance model predicts a 45-dimensional hand joint pose for $P$, which is retargeted to a 16-dimensional soft hand pose.

$$\mathcal{L} = \lambda_\mu ||\mu - \hat{\mu}||_2 + \lambda_\theta ||\theta_{\text{wrist}} - \hat{\theta}_{\text{wrist}}||_2 + \lambda_P ||P - \hat{P}||_2 \qquad (1)$$

At test time, we generate a crop of the object using Segment-Anything [71] and give our model a task description. The model generates contact points on the object, and we take the average as our contact point. Using a depth camera, we can determine the 3D contact point to navigate to. While the model outputs MANO parameters [30] that are designed to describe human hand joints, we retarget

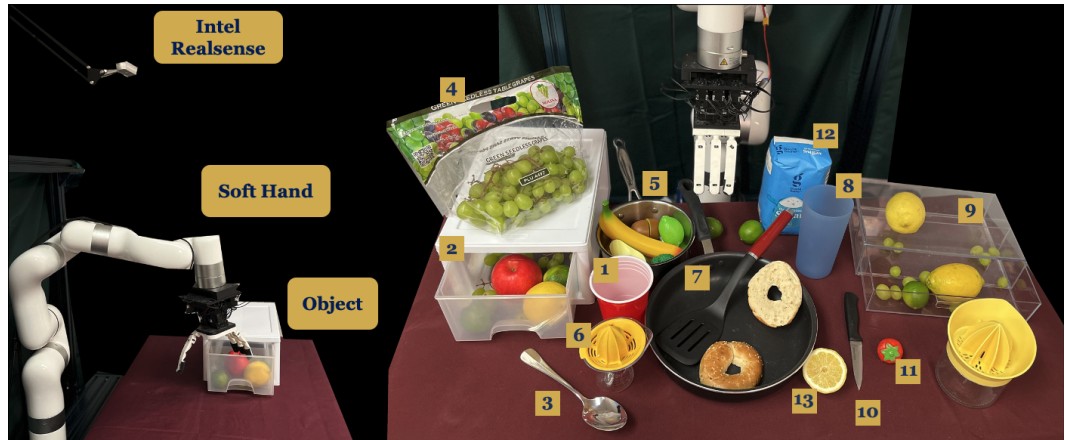

Figure 4: **Left**: Workspace Setup. We place an Intel RealSense camera above the robot to maintain an egocentric viewpoint, consistent with the affordance model's training data. **Right**: Thirteen objects used in our experiments.

these values to produce similar grasping poses on our robot hand in a similar manner to previous approaches [72, 48]. For more details, we refer readers to the appendix.

In addition to these grasp priors, we need a task-specific post-contact trajectory to successfully execute a task. Because it is challenging to learn complex and high-frequency action information from purely offline videos, we collect one demo of the human doing the robot task (Figure 3) separate from the affordance model $f$. We extract the task-specific wrist trajectory after the grasp using [34]. We compute the change in wrist pose between adjacent timesteps for the first 40 timesteps. When deployed for fine-tuning, we execute these displacements for the post-grasp trajectory. Once we have this prior, how can the robot *improve* upon it?

## 3.2   Fine-tuning via Interaction

---

**Algorithm 1** Fine-Tuning Procedure for DEFT

---

**Require:** Task-conditioned affordance model $f$, task description $T$, post-grasp trajectory $\tau$, parameter distribution $\mathcal{D}$, residual cVAE policy $\pi$. $E$ number of elites, $M$ number of warm-up episodes, $N$ total iterations.
  $\mathcal{D} \leftarrow \mathcal{N}(\mathbf{0}, \sigma^2)$
  **for** $k = 1 \ldots N$ **do**
    $I_{k,0} \leftarrow$ initial image
    $\xi_k \leftarrow f(I_{k,0}, T)$
    Sample $\epsilon_k \sim D$
    Execute grasp from $\xi_k + \epsilon_k$, then trajectory $\tau$
    Collect reward $R_k$; reset environment
    **if** $k > M$ **then**
      Order traj indices $i_1, i_2, \ldots, i_k$ based on rewards
      $\Omega \leftarrow \{\epsilon_{i_1}, \epsilon_{i_2}, \ldots, \epsilon_{i_E}\}$
      Fit $\mathcal{D}$ to distribution of residuals in $\Omega$
    **end if**
  **end for**
  Fit $\pi(.)$ as a VAE to $\Omega$

---

The affordance prior allows the robot to narrow down its learning behavior to a small subset of all possible behaviors. However, these affordances are not perfect and the robot will oftentimes still not complete the task. This is partially due to morphology differences between the human and robot hands, inaccurate detections of the human hands, or differences in the task setup. To improve upon the prior, we practice learning a residual policy for the grasp parameters in Table 1.

Residual policies have been used previously to efficiently explore in the real world [11, 73]. They use the prior as a starting point and explore nearby. Let the grasp location, wrist rotation, grasp pose, and trajectory from our affordance prior be $\xi$. During training we sample noise $\epsilon \sim \mathcal{D}$ where $\mathcal{D}$ is initialized to $\mathcal{N}(0, \sigma^2)$ (for a small $\sigma$). We rollout a trajectory parameterized by $\xi + \epsilon$. We collect $R_i$, the reward for each $\xi_i = f(v_i) + \epsilon_i$ where $v_i$ is the image. First, we execute an initial number of $M$ warmup episodes with actions sampled from $\mathcal{D}$, recording a reward $R_i$ based on how well the trajectory completes the task. For each episode afterward, we rank the prior episodes based on the reward $R_i$ and extract the sampled noise from the episodes with the highest reward (the 'elites' $\Omega$). We fit $\mathcal{D}$ to the elite episodes to improve the sampled noise. Then we sample actions from $\mathcal{D}$, execute the episode, and

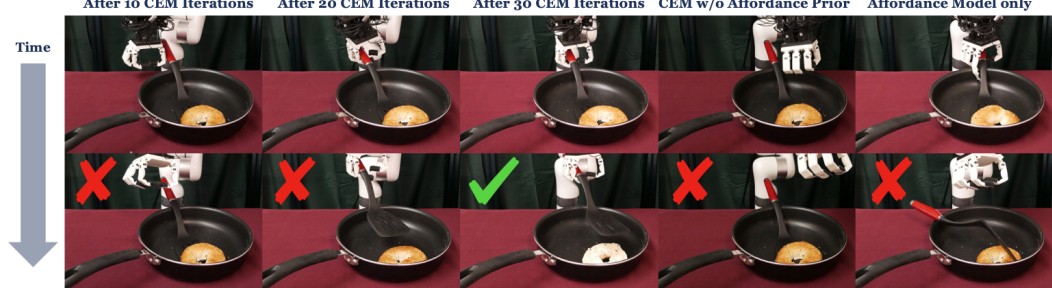

Figure 5: Qualitative results showing the finetuning procedure for DEFT. The model learns to hold the spatula and flip the bagel after 30 CEM iterations.

record the reward. By repeating this process we can gradually narrow the distribution around the desired values. In practice, we use $M = 10$ warmup episodes and a total of $N = 30$ episodes total for each task. This procedure is shown in Algorithm 1. See Table 4 for more information.

At test time, we could take the mean values of the top $N$ trajectories for the rollout policy. However, this does not account for the appearance of different objects, previously unseen object configurations, or other properties in the environment. To generalize to different initializations, we train a VAE [74, 75, 76, 77] to output residuals $\delta_j$ conditioned on an encoding of the initial image $\phi(I_{j,0})$ and affordance model outputs $\xi_j$ from the top ten trajectories. We train an encoder $q(z|\delta_j, c_j)$ where $c_j = (\phi(I_{j,0}), \xi_j)$, as well as a decoder $p(\delta_j|z, c_j)$ that learns to reconstruct residuals $\delta_j$. At test time, our residual policy $\pi(I_0, \xi)$ samples the latent $z \sim \mathcal{N}(\mathbf{0}, \mathbf{I})$ and predicts $\hat{\delta} = p(z, (I_0, \xi))$. Then we rollout the trajectory determined by the parameters $\xi + \hat{\delta}$. Because the VAE is conditioned on the initial image, we generalize to different locations and configurations of the object.

# 4   Experiment Setup

We perform a variety of experiments to answer the following: 1) How well can DEFT learn and improve in the real world? 2) How good is our affordance model? 3) How can the experience collected by DEFT be distilled into a policy? 4) How can DEFT be used for complex, soft object manipulation? Please see our website at http://dexterous-finetuning.github.io for videos.

**Task Setup**   We introduce 9 tabletop tasks, *Pick Cup*, *Pour Cup*, *Open Drawer*, *Pick Spoon*, *Scoop Grape*, *Stir Spoon*, *Pick Grape*, *Flip Bagel*, *Squeeze Lemon*. Robotic hands are especially well-suited for these tasks because most of them require holding curved objects or manipulating objects with tools to succeed. For all tasks, we randomize the position of the object on the table, as well as use train and test objects with different shapes and appearances to test for generalization. To achieve real-world learning with the soft robot hand, we pretrain an internet affordance model as a prior for robot behavior. As explained in Section 3, we train one language-conditioned model on all data. At test time, we use this as initialization for our real-world fine-tuning. The fine-tuning is done purely in the real world. An operator runs 10 warmup episodes of CEM, followed by 20 episodes that continually update the noise distribution, improving the policy. After this stage, we train a residual VAE policy that trains on the top ten CEM episodes to predict the noise given the image and affordance outputs. We evaluate how effectively the VAE predicts the residuals on each of the tasks by averaging over 10 trials. Because it takes less than an hour to fine-tune for one task, we are able to thoroughly evaluate our method on 9 tasks, involving over 100 hours of real-world data collection.

**Hardware Setup**   We use a 6-DOF UFactory xArm6 robot arm for all our experiments. We attach it to a 16-DOF Soft Hand using a custom, 3D-printed base. We use a single, egocentric RGBD camera to capture the 3D location of the object in the camera frame. We calibrate the camera so that the predictions of the affordance model can be converted to and executed in the robot frame. The flexibility of the robot hand also makes it robust to collisions with objects or unexpected contact with the environment. For the arm, we ensure that it stays above the tabletop. The job will be terminated if the arm's dynamics controller senses that the arm collided aggressively with the environment.

| Method | Pick cup | | Pour cup | | Open drawer | | Pick spoon | | Scoop Grape | | Stir Spoon | |
|---|---|---|---|---|---|---|---|---|---|---|---|---|
| | train | test | train | test | train | test | train | test | train | test | train | test |
| `Real-World Only` | 0.0 | 0.1 | 0.2 | 0.1 | 0.1 | 0.0 | 0.7 | 0.3 | 0.0 | 0.0 | 0.3 | 0.0 |
| `Affordance Model Only` | 0.1 | | 0.4 | | **0.5** | | 0.5 | | 0.0 | | 0.3 | |
| `DEFT` | **0.8** | **0.8** | **0.8** | **0.9** | **0.5** | **0.4** | **0.8** | **0.6** | **0.7** | **0.3** | **0.8** | **0.5** |

Table 2: We present the results of our method as well as compare them to other baselines: Real-world learning without internet priors used as guidance and the affordance model outputs without real-world learning.We evaluate the success of the methods on the tasks over 10 trials.

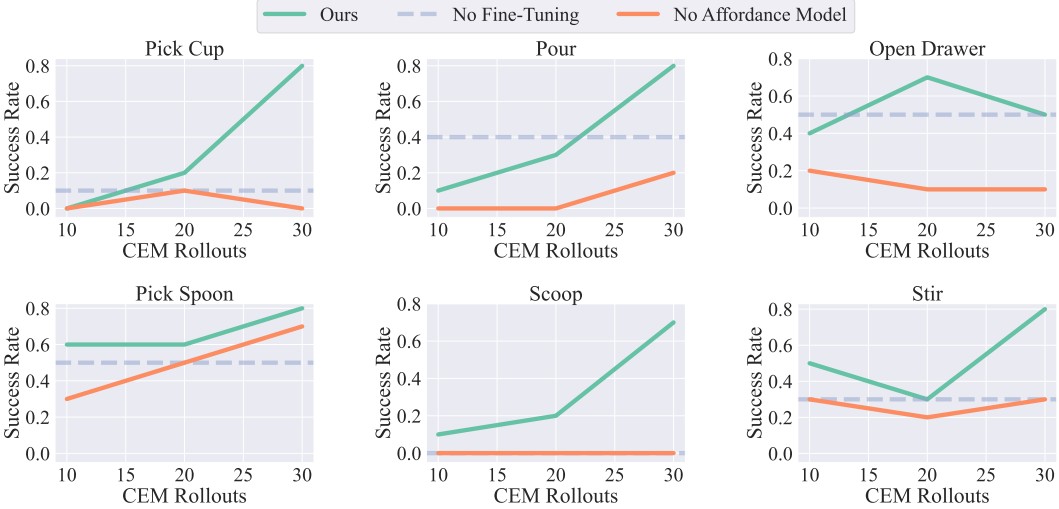

Figure 6: Improvement results for 6 tasks: pick cup, pour, open drawer, pick spoon, scoop, and stir. We see a steady improvement in our method as more CEM episodes are collected.

## 5 Results

**Effect of affordance model**  We investigate the role of the affordance model and real-world fine-tuning (Table 2 and Figure 6). In the real-world only model, we provide a few heuristics in place of the affordance prior. We detect the object in the scene using a popular object detection model [71] and let the contact location prior be the center of the bounding box. We randomly sample the rotation angle and use a half-closed hand as the grasp pose prior. With these manually provided priors, the robot has difficulty finding stable grasps. The main challenge was finding the correct rotation angle for the hand. Hand rotation is very important for many tool manipulation tasks because it requires not only picking the tool but also grasping in a stable manner.

**Zero-shot model execution**  We explore the zero-shot performance of our prior. Without applying any online fine-tuning to our affordance model, we rollout the trajectory parameterized by the prior. While our model is decent on simpler tasks, the model struggles on tasks like stir and scoop that require strong power grasps (shown in Table 2). In these tasks, the spoon collides with other objects, so fine-tuning the prior to hold the back of the spoon is important in maintaining a reliable grip throughout the post-grasp motion. Because DEFT incorporates real-world experience with the prior, it is able to sample contact locations and grasp rotations that can better execute the task.

**Human and automated rewards**  We ablate the reward function used to evaluate episodes. Our method queries the operator during the task reset process to assign a continuous score from 0 to 1 for the grasp. Because the reset process requires a human-in-the-loop regardless, this adds little marginal cost for the operator. But what if we would like these rewards to be calculated autonomously? We use the final image collected in the single post-grasp human demonstration from Section 3 as the goal image. We define the reward to be the negative embedding distance between the final image of the episode and the goal image with either an R3M [69] or a ResNet [78] encoder. The model learned from ranking trajectories with R3M reward is competitive with DEFT in all but one task, indicating that using a visual reward model can provide reasonable results compared to human rewards.

| Method | Pour Cup | | Open Drawer | | Pick Spoon | |
|---|---|---|---|---|---|---|
| | train | test | train | test | train | test |
| *Reward Function:* | | | | | | |
| `R3M Reward` | 0.0 | 0.0 | 0.4 | **0.5** | 0.5 | 0.4 |
| `Resnet18 Imagenet Reward` | 0.1 | 0.2 | 0.3 | 0.1 | 0.4 | 0.2 |
| *Policy Ablation:* | | | | | | |
| `DEFT w/ MLP` | 0.0 | 0.0 | 0.5 | 0.0 | 0.6 | 0.5 |
| `DEFT w/ Transformer` | 0.4 | 0.5 | **0.6** | 0.1 | 0.4 | 0.5 |
| `DEFT w/ Direct Parameter est.` | 0.1 | 0.1 | 0.1 | 0.0 | 0.3 | 0.0 |
| `DEFT` | **0.8** | **0.9** | 0.5 | 0.4 | **0.8** | **0.6** |

Table 3: Ablations for (1) reward function type, (2) model architecture, and (3) parameter estimation.

**Model Architecture** We investigate different models and training architectures for the policy trained on the rollouts (Table 3). When we replace the conditional VAE with an MLP that predicts residuals, the model has dif-

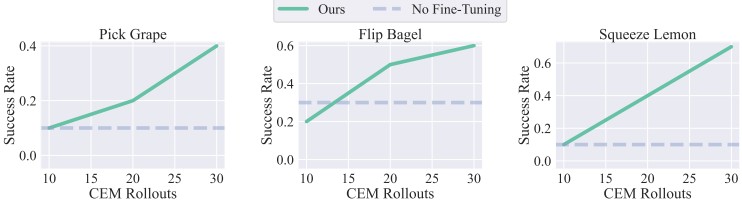

Figure 7: We evaluate DEFT on three difficult manipulation tasks.

ficulty learning the grasp rotation to effectively pour a cup. We find that the MLP cannot learn the multi-modality of the successful data properly. Our transformer ablation is an offline method similar to [79] where in addition to the image and affordance model outputs, we condition on the reward outputs and train a transformer to predict the residual. At test time the maximum reward is queried and the output is used in the rollout. While this method performs well, we hypothesize that the transformer needs more data to match DEFT. Finally, we train a VAE to directly estimate $\xi$ instead of the residual. This does not effectively distill the information from the affordance prior without the training time allotted. As a result, it often makes predictions that are far from the correct grasp pose.

**Performance on complex tasks and soft object manipulation** We investigate the performance of DEFT on more challenging tasks. Tasks involving soft objects cannot be simulated accurately, while our method is able to perform reasonably on food manipulation tasks as shown in Figure 7.

Of the three tasks, our method has the most difficulty with the Pick Grape task. Because grapes are small, the fingers must curl fully to maintain a stable grasp. A limitation of our hand is that the range of its joints does not allow it to close the grasp fully and as a result, it has difficulty in consistently picking small objects. This also makes it challenging to hold heavy objects like the spatula in Flip Bagel, but with practice DEFT learns to maintain a stable grasp of the spatula. For Squeeze Lemon, DEFT develops a grasp that allows it to apply sufficient pressure above the juicer. Specifically, our method takes advantage of the additional fingers available for support in hands.

## 6 Discussion and Limitations

In this paper, we investigate how to learn dexterous manipulation in complex setups. DEFT aims to learn directly in the real world. In order to accelerate real-world fine-tuning, we build an *affordance* prior learned from human videos. We are able to efficiently practice and improve in the real world via our online fine-tuning approach with a soft anthropomorphic hand, performing a variety of tasks (involving both rigid and soft objects). While our method shows some success on these tasks, there are some limitations to DEFT that hinder its efficacy. Although we are able to learn policies for the high-dimensional robot hand, the grasps learned are not very multi-modal and do not capture all of the different grasps humans are able to perform. This is mainly due to noisy hand detections in affordance pretraining. As detection models improve, we hope to be able to learn a more diverse set of hand grasps. Second, during finetuning, resets require human input and intervention. This limits the real-world learning we can do, as the human has to be constantly in the loop to reset the objects. Lastly, the hand's fingers cannot curl fully. This physical limitation makes it difficult to hold thin objects tightly. Future iterations of the soft hand can be designed to grip such objects strongly.

## Acknowledgments

We thank Ananye Agarwal and Shagun Uppal for fruitful discussions. KS is supported by the NSF Graduate Research Fellowship under Grant No. DGE2140739. The work is supported in part by ONR N00014-22-1-2096, ONR MURI N00014-22-1-2773, and Air Force Office of Scientific Research (AFOSR) FA9550-23-1-0747.

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

## A    Video Demo

We provide video demos of our system at `https://dexterous-finetuning.github.io`.

## B    DASH: Dexterous Anthropomorphic Soft Hand

Recently introduced, DASH (Dexterous Anthropomorphic Soft Hand) [1] is a four-fingered anthropomorphic soft robotic hand well-suited for machine learning research use. Its human-like size and form factor allow us to retarget human hand grasps to robot hand grasps easily as well as perform human-like grasps. Each finger is actuated by 3 motors connected to string-like tendons, which deform the joints closest to the fingertip (DIP joint), the middle joint (PIP joint), and the joint at the base of the finger (MCP joint). There is one motor for the finger to move side-to-side at the MCP joint, one for the finger to move forward at the MCP joint, and one for PIP and DIP joints. The PIP and DIP joints are coupled to one motor and move dependently. While the motors do not know the end-effector positions of the fingers, we learn a mapping function from pairs of motor angles and visually observed open-loop finger joint angles. These models are used to command the finger joint positions learned from human grasps.

## C    MANO Retargeting

For MANO parameters, the axis of each of the joints is rotation aligned with the wrist joint and translated across the hand. However, our robot hand operates on forward and side-to-side joint angles. To translate the MANO parameters to the robot fingers we extract the anatomical consistent axes of MANO using MANOTorch. Once these axes are extracted, each axis rotation represents twisting (not possible for human hands), bending, and spreading. We then match these axes to the robot hand. The spreading of the human hand's fingers (side-to-side motion at the MCP joint) maps to the side-to-side motion at the robot hand's base joint. The forward folding at the base of the human hand (forward motion at the MCP joint) maps to the forward motion at the base of the robot hand's finger. Finally, the bending of the other two finger joints on the human hand, PIP and DIP, map to the robot hand's PIP and DIP joints. While the thumb does not have anatomically the same structure, we map the axes in the same way. Other approaches rely on creating an energy function to map the human hand to the robot hand. However, because the soft hand is similar in anatomy and size to a human hand, it does not require energy functions for accurate retargeting.

## D    Affordance Model Training

We use data from Ego4D [42], EpicKitchens-100 [44], and HOI4D [8]. After filtering for clips of sufficient length, clips that involve grasping objects with the right hand, and clips that have language annotations, we used 64666 clips from Ego4D, 9144 clips from EpicKitchens, and 2707 clips from HOI4D. In total, we use a dataset of 76517 samples for training our model.

For our contact location model, we use the visual encoder from [69] to encode the image as a 512-dimensional vector. We use the spatial features of the encoder to upsample the latent before applying a spatial softmax to return the contact heatmap. This consists of three deconvolutional layers with 512, 256, and 64 channels in that order.

To predict wrist rotation and grasp pose, we use the language encoder from [70] to compress the language instruction to a 512-dimensional vector. We concatenate the visual and language latents and pass them through a transformer with eight heads and six self-attention layers. We pass the result of the transformer through an MLP with hidden size 576, and predict a vector of size 48: the first 3 dimensions are the axis-angle rotations; the last 45 dimensions are the joint angles of the hand. These correspond to the parameters output by Frankmocap [34], which we used to get ground truth hand pose in all the datasets. The images used from the training datasets as well as the ground truth labels are released here.

We jointly optimize the L2 loss of the contact location $\mu$, the wrist rotation $\theta_{\text{wrist}}$ and grasp pose $P$. The weights we used for the losses are $\lambda_\mu = 1.0, \lambda_\theta = 0.1, \lambda_P = 0.1$. We train for 70 epochs with an initial learning rate of 0.0002, and a batch size of 224. We used the Adam optimizer [80] with cosine learning rate scheduler. We trained on a single NVIDIA RTX A6000 with 48GB RAM.

## E  Fine-Tuning Parameters

Below are the values of the parameters used for the CEM phase of DEFT.

| Parameter | Value | Description |
|---|---|---|
| $E$ | 10 | Number of elites for CEM |
| $M$ | 10 | Number of warm-up episodes |
| $N$ | 30 | Total number of CEM episodes |
| $\sigma_\mu$ | 0.02 | Initial contact location Standard Deviation (meters) |
| $\sigma_{\theta_{\text{wrist}}}$ | 0.2 | Initial wrist rotation Standard Deviation (euler angle radians) |
| $\sigma_P$ | 0.05 | Initial soft hand joints Standard Deviation |

Table 4: Values for fixed parameters in fine-tuning Algorithm 1.

## F  Success Criteria for Tasks

We define the criteria for success in each of our 9 tasks as follows:

- Pick Cup: Cup must leave table surface and stay grasped throughout trial.
- Pour Cup: Cup must be grasped throughout trial and also rotate so that the top of the cup is at a lower height than the base.
- Open Drawer: Drawer is initially slightly open so that it can be grasped. By the end of the episodes, the drawer should be at least 1 centimeter more open than it was at the beginning.
- Pick Spoon: The spoon must not be in contact with the table at the end of the trial.
- Stir Spoon: The spoon base must rotate around the jar/pot at least 180 degrees while grasped.
- Scoop Grape: The spoon must hold a grape at the end of the trial while being held by the soft hand.
- Pick Grape: All grapes must be held by the hand above the table surface. In particular, if any single grape falls due to a weak stem, this is considered a failure.
- Flip Bagel: The side of the bagel that is facing up at the end of the trial should be opposite the side facing up at the beginning.
- Squeeze Lemon: The lemon should be grasped securely on top of the juicer.

