# OpenReview forum: "DEFT: Dexterous Fine-Tuning for Hand Policies"
_robot-learning.org/CoRL/2023/Conference — CoRL 2023 Poster_

### Official Review · Reviewer_Tfey · 2023-07-18

**Confidence:** 3
**Originality:** Good
**Technical Quality:** Good
**Clarity Of Presentation:** Fair
**Impact:** 3

**Recommendation:**

Weak Accept: I recommend accepting the paper, but will not argue for my recommendation if the majority of other reviewers have a different opinion.

**Review:**

Strengths:
- the paper addresses a very relevant problem in robotics
- paper proposes an interesting overall pipeline

Weaknesses:
- the paper lacks clarity in quite some locations, i.e., many crucial details of the method are unclear, and also no further details are provided in the appendix (i.e., dimensions of variables, number of training data, number of samples for evaluation, etc.). It thus feels that it would be impossible to reproduce the results given the current state of the paper. That said, it also feels that the appendix is incomplete.
- I believe the experiments are missing some important and relevant ablations. I.e., how much does the affordance prior change / react to the text task description? I.e., is the text description always something like "grasp object x" or does it also contain information about the task that has to be completed after grasping the object, i.e., flip the object?
- I was unclear why the human wrist trajectories could not be recovered from the video data. Why was one explicit human demonstration needed for the post-grasp trajectory?
- the paper would substantially improve its impact if the code and data for training the models, as well as the data from the experiments would be open-sourced


This paper investigates a relevant problem in robotics, achieving dexterous manipulation using complicated, high-dimensional hardware, i.e., robotic hands.

Generally speaking, the introduction section could be made more concise. Moreover, some statements in the introduction could be underlined with citations. Also, there exist better arguments why we need multi-fingered hands for dexterous robotic manipulation than tying shoelaces. For the latter case, I would argue that a bimanual manipulator is inevitable and perhaps more important.

Additionally, the authors should better highlight in the related work section how this work differs from existing works and where the contributions are. This, in particular, holds true for the section on soft object manipulation, where I feel that the authors should comment or at least briefly mention how they actually achieve soft object manipulation, as they do not use touch sensors or direct force feedback, but rather soft, compliant hardware.

I also feel that section 3 does not contain enough information for it to be a section on its own. Therefore, I propose adding this information at the beginning of the method section.

Regarding the method, in my opinion, the authors might need to create a better overview figure depicting the individual components. While Figure 2 goes in this direction, it is not referenced in the text and thus might be overlooked in the current version of the manuscript.
More critically, the method section is missing crucial details regarding the method. In particular, I would appreciate it if the authors added the information in which spaces the individual components live. E.g., do the contact points $\mu$ with centers $\mu_1 ... \mu_k$ live in the space of pixel coordinates, or are these already 3D coordinates. Second, how many dimensions does the hand grasp pose P have? Third, is the wrist orientation $\theta_{wrist}$ a single scalar or actually a 3D rotation? Also, I was generally wondering why the contact points mu have to be estimated at all. Would not the hand grasp pose P and the wrist orientation be sufficient?
Lastly, the post-grasp trajectory parameterization was also unclear to me. Is it defined as a set of waypoints (if yes, how many?) or differently? Moreover, as mentioned previously, it would be interesting to know about the versatility of the text labels, and it would be even more interesting to see experiments on whether the task description actually influences the grasp pose when trying to achieve different tasks using the same tool.
While it is possible to recover some of this information from the related and cited works, it is absolutely essential that the authors provide all this information in the paper.

Moreover, the authors mention that they extract affordances from three large-scale egocentric datasets. It would here again be interesting whether the authors trained on all the data from the datasets or whether they focussed on specific samples / scenarios that are related to the settings presented in the paper. Ideally, the authors would release the data that they trained their affordance model on with the paper.

Moreover, the authors mention in the methods section that for the post-grasp movement they "collect one demo of the human doing the task". As for instance the videos of the EPIC kitchens dataset, in theory, also contain the trajectory of the hand to complete the task, I was wondering why this information has to be provided separately and is not extracted from the dataset. It would be great if the authors could comment on that.

In line with the question about the dimensionality of the method's components, I was also wondering about the dimensionality of the search space of the sampling-based optimization that is introduced in Section 4.2. Moreover, it would also be good to know how the authors decided on the initial noise scale for the sampling-based optimization. They also do not provide any information on how many trajectories / rollouts are done per episode.

In the results section, it was again unfortunately unclear to me how the success percentages have been obtained. I assume that they are the average of repeating the task x times, however, this information is not specified in the paper.

Moreover, in Table I, the authors differentiate between train and test scenarios. What was unclear to me is whether the "train" scenarios relate to evaluating the final noise distribution obtained by the sampling-based CEM optimization or whether "train" relates to evaluating the trained VAE simply in the same scenario that it was trained in. Depending on this information, the drop in performance between train and test that can be observed for DEFT could either be due to the change in the scene or due to the VAE having problems learning the residuals. It would thus be important to have this kind of information.
Additionally, while the authors mention that they evaluate the method in different scenarios compared to training, it would be nice if they could provide pictures / videos that showcase how different the scenarios are.

Next, it felt to me that the author's statement at the beginning of the experiment setup section, i.e., "over all our experiments, we collect data for several thousands of rollouts for over 100 hours of real-world data collection" somehow even seems to contradict with the statement in the introduction of "the method enables iterative real-world improvement in less than one hour". It would thus be great if the authors could differentiate between the timely effort to refine the policies on all tasks and the time needed for performing the thorough evaluations.

Finally, I really appreciated the honest statement at the end of the introduction (i.e., "While our method begins to show good success on these tasks ... more evaluation is required to complete these tasks more effectively"). Nevertheless, a similar statement should be added in the actual limitations and discussion section.

**Quality Of The Limitations Section:**

Additional details required

**Questions For Rebuttal:**

- many details regarding the method need to be clarified, as otherwise it is impossible to reproduce the method and the experimental results
- would it be possible to open source the code as well as the affordance dataset? This would heavily benefit the paper
- why is an additional demonstration needed for the post-grasp movement?
- it would be great if the authors could provide additional details on whether the text input to the affordance model actually impacts the grasping pose

**Robotics Focus:**

Sufficient demonstration on hardware

**Summary Of Paper:**

This paper proposes an approach for achieving dexterous manipulation using a high dimensional robotic hand. The approach is based on fine-tuning manipulation behavior. The initial manipulation trajectory is obtained through a learned affordance model that is trained on video data and essentially handles grasp generation, together with one human demo of the task for obtaining the post-grasping movement.
This initial manipulation trajectory is then optimized by performing sampling-based optimization (CEM-like) on the real system. The result of the online optimization using CEM is subsequently distilled into a residual variational autoencoder. The VAE allows directly estimating residual terms when facing a previously unseen and slightly modified testing environment.
The authors subsequently present an experimental evaluation of their method, illustrating the effect of policy fine-tuning, as well as presenting ablations for different reward functions and ablations of the policy.

**Summary Of Recommendation:**

The paper addresses a relevant robotics problem and introduces an interesting pipeline. However, in its current form, it is missing essential details in almost all of the sections, and some points need to be clarified to understand its potential impact. I, therefore, currently lean towards rejecting the paper.

---

### Official Review · Reviewer_g2GV · 2023-07-18

**Confidence:** 3
**Originality:** Fair
**Technical Quality:** Fair
**Clarity Of Presentation:** Very Good
**Impact:** 3

**Recommendation:**

Weak Accept: I recommend accepting the paper, but will not argue for my recommendation if the majority of other reviewers have a different opinion.

**Review:**

Strengths:
- The proposed method is very sample efficient, and can fine-tune in under an hour in the real-world.
- The method showcases impressive results in some of the tasks; most notably the “Pour Cup” task which highlights the advantages of an anthropomorphic hand.
- The majority of the technical details are well-specified and easy to follow.
- The visualized rollouts are very helpful for qualitatively evaluating the resulting learned policy.


Weaknesses:
- I am not convinced that the proposed task suite is most suitable to showcase the advantages of using an anthropomorphic hand. With the exception of “Pour Cup” or “Squeeze Lemon”, one could argue that all of the tasks could be completed with a parallel jaw gripper, and perhaps with greater stability as well. Justification of the tasks chosen would help clarify the scope of the proposed method and strengthen the overall argument.
- A central argument of the paper is that the proposed method improves anthropomorphic dexterity; however, the visualized rollouts do not seem to suggest that the robot consistently learns stable grasps. In many tasks, such as “Flip Bagel”, “Stir Spoon”, and “Pick Spoon”, the grasped object looks as if it is on the verge of dropping, and in many cases some of the fingers seem to have no impact on the resulting grasp. This may be a limitation of the specific hardware used, but in any case it would helpful to explicitly address these obvious visual discrepancies.
- Currently, it is unclear how DEFT is expected to perform on more complex tasks involving soft object manipulation. While there is a brief discussion in L253-L255, further discussion may be necessary – as that section stands, it does not provide any concrete insight into DEFT’s performance under these settings.

**Quality Of The Limitations Section:**

Limitations are addressed clearly

**Questions For Rebuttal:**

Questions:
- How is success determined? Based on the visualized rollouts, it does not seem clear to me that some tasks (e.g.: “Stir Spoon” or “Squeeze Lemon” tasks) are actually being completed, at least semantically.
- It seems the crux of the problem is effectively executing a given grasp and maintaining grasp stability throughout a desired trajectory. Is DEFT expected to be robust to unstable grasps or is this a limitation of the method?
- What is the motivation for selecting these specific 9 tasks? How does these specific tasks showcase both the advantages of an anthropomorphic hand as well as the strength of the proposed method?

Technical Concerns:
- L208: Syntax errors / typos
- The 9 listed tasks in L192-L193 and Table 1 do not match the tasks shown in Figure 1. I assume “Scoop Sugar” and “Stir Straw” are outdated?
- Scoop Grape task missing from website

**Robotics Focus:**

Sufficient demonstration on hardware

**Summary Of Paper:**

The paper proposes a novel method (DEFT) that leverages human-driven priors to quickly learn grasping policies that can immediately be deployed in the real world on an anthropomorphic robot hand. The method is evaluated on 9 real-world tasks, and is shown to outperform baselines that do not include either the real-world finetuning or affordance model.

**Summary Of Recommendation:**

The work proposes a novel method (DEFT) that can be trained and finetuned quickly in the real world, and showcases impressive results on certain real-world tasks, including the “Pour Cup” task involving liquids. However, the experimental setup and evaluation method lacks some key details and motivation, and further discussion on the intended scope of the method is needed.

---

### Official Review · Reviewer_1dz2 · 2023-07-20

**Confidence:** 4
**Originality:** Good
**Technical Quality:** Good
**Clarity Of Presentation:** Good
**Impact:** 3

**Recommendation:**

Weak Accept: I recommend accepting the paper, but will not argue for my recommendation if the majority of other reviewers have a different opinion.

**Review:**

Strength:

- The method combines the prior knowledge extracted from the human videos and the online data for fine-tuning, achieving dexterous manipulation in the real world. I appreciate real-world experiments. The result looks impressive to me, and the method is quite promising. I was not aware such simple CEM methods were applied in the real world. I think the paper contains sufficient contribution for acceptance.

Weakness:

- However, despite the authors' efforts to build and implement the system, I found essential details missing in the paper, making it hard to provide a fair evaluation. I tried to find the details in the appendix as promised in Line 185 of the main text, which confused me about the CEM and VAE part. What are the differences between the “rollouts” and “episodes”? Are they the same concepts? How do you train the VAE with only 10 data samples? Won’t this cause overfitting? How many rollouts are required for each task? I hope the authors can provide more details regarding the implementation.
- I wonder if CEM is the best choice for online-finetuning. Before the details are clarified, I am worried that online fine-tuning will cost too much or can not generalize well to different initialization. Methods like iterative residual policy sound more data-efficient than the CEM and may benefit from large-scale video datasets. I’d hear the authors’ thoughts on this.
- The method contains a complex pipeline for extracting suitable priors for grasping, but I need to see ablation studies regarding the visual parts. The authors should provide more analysis details, at least in the appendix, to help readers better understand its contribution.

**Quality Of The Limitations Section:**

Limitations are addressed clearly

**Questions For Rebuttal:**

- please provide the details for CEM and VAE training.

**Robotics Focus:**

Sufficient demonstration on hardware

**Summary Of Paper:**

The authors present DEFT, a method that learns useful dexterous manipulation in the real world that can generalize to many objects and scenarios. It first learns prior affordances from human videos and fits a transformer to jointly predict the pose, orientation, and affordance. It then finetunes the policies extracted from human videos through online interaction by a CEM-like trajectory optimization method in the real world. As a result it can control dexterous hands to solve 9 real-world tasks with limited online interactions.

**Summary Of Recommendation:**

I think the paper presents a nice system for real-world dexterous hand manipulation. I appreciate the authors’ effort but I feel details are missing now and need clarification before acceptance.

---

### Author Response · Authors · 2023-08-16
**Note to AC: Summary and Updates**

Dear AC,

We thank the reviewers for the insightful feedback on our paper. We are glad that the reviewers appreciated our real world experiments (R #g2GV, R #1dz2), had impressive real world results (R #g2GV, R #1dz2), addressed “a very relevant problem in robotics” (R #Tfey) and “had sufficient contribution for acceptance” (R #1dz2).

We are pleased to report that we have addressed conceptual and experimental clarifications that reviewers asked for. To this end, Reviewer #Tfey has recommended acceptance and said that they would increase the score. Unfortunately, we didn’t receive a reply from two of the thre reviewers in time for the discussion period for us to respond further but we hope to have addressed all their concerns.

We have made significant improvements to improve the paper's overall clarity by:

(1) Delineating the steps of the fine-tuning process in a pseudocode box, (2) adding tables that specify the method parameters so that it can be easily reproduced, (3) better motivating our method and use of robotic hands, (4) providing additional analysis for challenging tasks, and (5) clarifying the physical limitations of our system.


Thank you!

Authors

---

### Decision · Program_Chairs · 2023-08-30

**Decision:**

Accept (Poster)

**Comment:**

The authors propose a method to learn policies for a high-dof anthropomorphic hand.  The method has a couple key ingredients including learning affordances from videos of humans, and a CEM-type optimization method for finetuning the robot policy.

The reviews for this paper had considerable detail, and although all 3 reviewers started at weak reject, all 3 reviewers updated to weak accept after the rebuttal.  In initial reviews, reviewers pointed to various strengths of the method, including the ability to use prior knowledge from human videos, addressing high-dof dexterous manipulation, and reasonably data-efficient real-robot behavior.  A common weakness area pointed out was the omission of key details, including on baseline comparisons and ablations concerning the effect of the affordance prior.  Reviewer g2GV also pointed out that many of these tasks could have been done with a parallel jaw gripper, which perhaps did not best highlight the capabilities of the hand.  The authors had robust discussion with the reviewers, and all reviewers updated to weak accept after understanding the answers to their questions.  Reviewers found the responses satisfactory given the constraints of rebuttal time, and of the used robot hardware.

The topic is very relevant to core problems of robot learning, and the results are encouraging and would be of considerable value for the community to see at CoRL.

As requested by reviewer Tfey and promised by the authors, the authors should release the datasets used for training. Also, it would certainly be nice for the authors to open source the code.  Also, as a minor suggestion to consider for future, the 6dof motion of the arm end-effector appears a bit jerky, and might be helped by having some high-rate smoothing and/or interpolation between different commands chosen by the learned robot policy.